# Sixteen isostructural phosphonate metal-organic frameworks with controlled Lewis acidity and chemical stability for asymmetric catalysis

Xu Chen[1], Yongwu Peng[1], Xing Han[1], Yan Liu[1], Xiaochao Lin[1] & Yong Cui [1,2]

Heterogeneous catalysts typically lack the specific steric control and rational electronic tuning required for precise asymmetric catalysis. Here we demonstrate that a phosphonate metal–organic framework (MOF) platform that is robust enough to accommodate up to 16 different metal clusters, allowing for systematic tuning of Lewis acidity, catalytic activity and enantioselectivity. A total of 16 chiral porous MOFs, with the framework formula $[M_3L_2(solvent)_2]$ that have the same channel structures but different surface-isolated Lewis acid metal sites, are prepared from a single phosphono-carboxylate ligand of 1,1′-biphenol and 16 different metal ions. The phosphonate MOFs possessing *tert*-butyl-coated channels exhibited high thermal stability and good tolerances to boiling water, weak acid and base. The MOFs provide a versatile family of heterogeneous catalysts for asymmetric allylboration, propargylation, Friedel–Crafts alkylation and sulfoxidation with good to high enantioselectivity. In contrast, the homogeneous catalyst systems cannot catalyze the test reactions enantioselectively.

[1] School of Chemistry and Chemical Engineering and State Key Laboratory of Metal Matrix Composites, Shanghai Jiao Tong University, Shanghai 200240, China. [2] Collaborative Innovation Center of Chemical Science and Engineering, Tianjin 300072, China. Xu Chen and Yongwu Peng contributed equally to this work. Correspondence and requests for materials should be addressed to Y.L. (email: liuy@sjtu.edu.cn) or to Y.C. (email: yongcui@sjtu.edu.cn)

Heterogeneous catalysts have a pervasive and indispensible role in industrial processes used to produce many essential chemicals and fuels. However, conventional heterogeneous catalysts lack the fine steric and electronic tuning required, especially for asymmetric catalysis of organic transformations, which provides enantiomerically enriched products of both academic and industrial interest still catalyzed by homogeneous catalysts[1,2]. Metal–organic frameworks (MOFs) have emerged as a leading class of porous crystalline materials for their rich structural architectures and various potential applications[3,4]. Because of the readily modification of their metal clusters (SBUs) and organic linkers, MOFs offer a useful platform for designing solid catalysts for organic transformations[5–9], especially for asymmetric reactions that could not be realized with traditional porous inorganic solids[10–27]. A variety of asymmetric MOF catalysts have been designed with privileged chiral ligands/catalysts such as BINOL- and metallosalen-based derivatives, but they are typically less effective, with limited substrate scope, than their homogeneous analogs[14–23]. The SBUs behave as unique supramolecular ligands for transition metal ions that are known to act as Lewis acid catalysts, providing an intriguing platform for other types of heterogeneous catalysis. Although the SBUs in MOFs have been explored for asymmetric catalysis[24–27], but satisfactory enantioselectivity was only achieved in one instance[26]. Another challenge facing MOF catalysts is their generally low stability to harsh reaction conditions such as boiling water, weak acid, and alkaline media[6,10], thus limiting their use in practical processes[28,29]. In this work, we demonstrated how to address such issues by designing a stable phosphonate MOF platform that can accommodate up to 16 different homometallic clusters as SBUs, allowing for systematic tuning of Lewis acidity, catalytic activity and enantioselectivity.

Phosphonates including arylphosphonates and arylphosphocarboxylates form stronger bonds with metal ions than pure carboxylates that have been explored for the synthesis of stable MOFs[30–33], but they tend to form dense layered crystalline materials or amorphous solids[31–33]. Chiral phosphoric acids of biaryl and their metal salts have emerged as powerful Brønsted acid or Leiws acid/Brønsted base catalysts in a range of enantioselective reactions because of their unique structural and chemical features[34]. Here we report a strategy of combining bulky hydrophobic groups and chiral metal phosphonate catalysts to make stable MOFs as versatile asymmetric catalysts. The premise of this approach was that a chiral phosphonate group would moderate the self-assembly and allow for stable yet crystalline MOF with metal nodes as chiral Lewis acids, while the hydrophobic groups could not only sterically shield the vulnerable M−O bonds and augment the hydrolytic stability of networks but also exert stereochemical and electronic control over catalytic reactions. For this purpose, we synthesized a chiral dicarboxylate linker derived from 1,1′-biphenol phosphoric acid with pendant *tert*-butyl group at the 3,3′-position that coordinates with metal

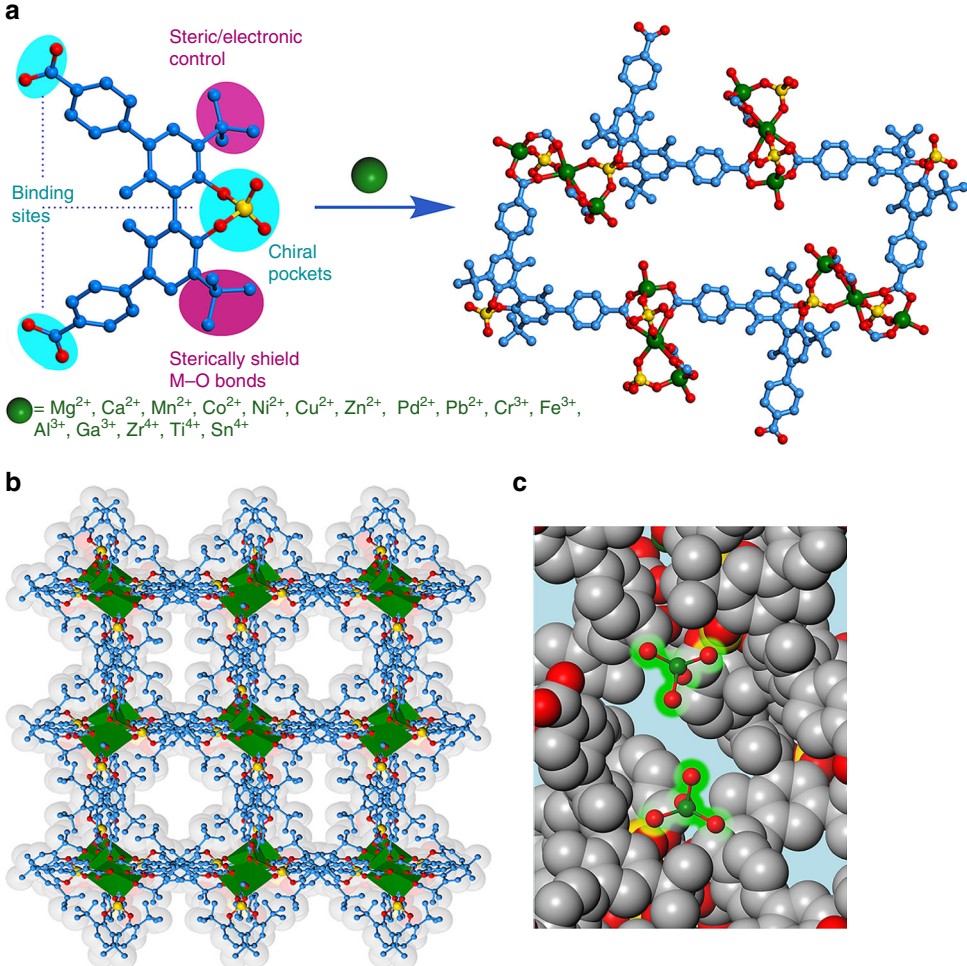

**Fig. 1** Synthesis and X-ray crystal structure of **1-M**. **a** Synthesis of the isostructural series of **1-M** from H₃**L** and different metal ions (only the coordinated oxygen atoms were shown for OAc⁻ anions in **1-M**(III) and **1-M**(IV). Green, M; gold, P; red, O; blue, C). **b** The 3D hydrophobic structure of **1-M** along the *c*-axis (the metal ions are shown in polyhedral). **c** A portion of the 3D structure highlighting the exposure of the SBUs to the pores

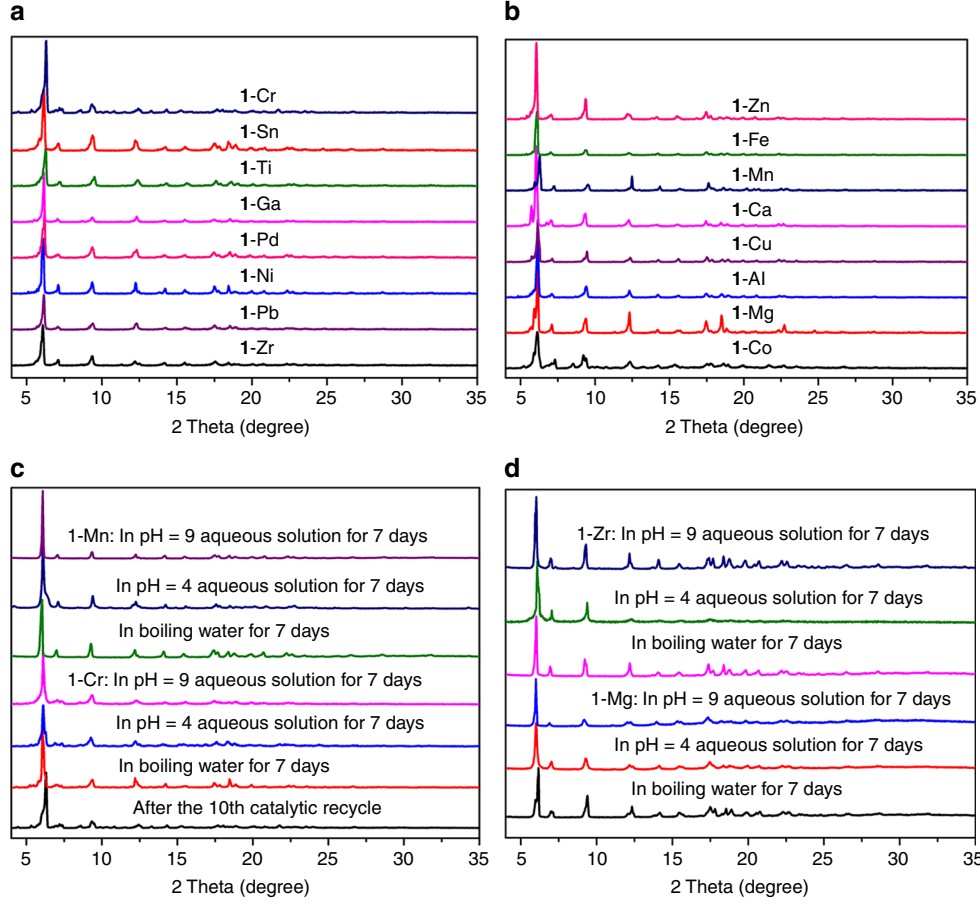

**Fig. 2** PXRD patterns of **1-M**. **a**, **b** The as-prepared **1-M**. **c** **1-Cr** and **1-Mn** after 7 days treatment under different conditions. **d** **1-Zr** and **1-Mg** after 7 days treatment under different conditions

ions via carboxylate and phosphonate groups to form a total of 16 isostructural porous MOFs bearing metal nodes as chiral Lewis acids. These phosphonate MOFs exhibit highly thermal stabilities and good tolerances to water, acids and bases. The MOFs are shown to be efficient heterogeneous catalysts for asymmetric allylboration, propargylation, Friedel–Crafts alkylation, and sulfoxidation with up to 99% ee, whereas the corresponding homogeneous catalysts cannot promote the reactions enantioselectively. The synthetic utilities of the MOF catalysts are demonstrated in the preparation of important building blocks of biological active compounds.

## Results

**Synthesis and X-ray structure of the MOFs.** The ligand (*S* or *R*)-3,3′-di-*tert*-butyl-5,5′-dicarboxyphenyl-6,6′-dimethylbiphenyl-2,2′-diylhydrogen phosphate (H₃**L**) was synthesized by a Pd-catalyzed Suzuki cross-coupling between enantiopure 3,3′-di-*tert*-butyl-5,5′-dibromo-6,6′-dimethylbiphenyl-2,2′-diol and 4-(methoxycarbonyl)-phenylboronic acid, followed by phosphorylation and base-catalyzed hydrolysis. Heating a mixture of metal salts and H₃**L** in a mixed solvents containing acetic acid at 80 °C for 1 day afforded block crystals of $[M_3L_2(H_2O)_2]$·guest (**1-M**) for $M^{2+} = Mg^{2+}, Ca^{2+}, Mn^{2+}, Co^{2+}, Ni^{2+}, Cu^{2+}, Zn^{2+}, Pb^{2+}$, and $Pd^{2+}$, $[M_3L_2(OAc)(H_2O)][OAc]_2$·guest for $M^{3+} = Cr^{3+}, Fe^{3+}, Al^{3+}$, and $Ga^{3+}$, and $[M_3L_2(OAc)(H_2O)][OAc]_2[OH]_3$·guest for $M^{4+} = Zr^{4+}, Ti^{4+}$, and $Sn^{4+}$ (Fig. 1). Five framework structures were determined by single-crystal X-ray diffraction (XRD).

**1-Cr** crystallizes in the chiral orthorhombic space group $C222_1$ and has a 3D network composed of three independent metal ions and two independent, fully deprotonated **L** ligands in a 2:3 molar ratio. The basic building block is a linear trimeric $Cr_3$ unit, which is linked by four bidentate carboxylate groups and two bidentate phosphonate groups of eight **L** ligands (Fig. 1). The central Cr is coordinated octahedrally by six oxygen atoms from four bidentate carboxylate and two bidentate phosphonate groups, and each terminal Cr is coordinated tetrahedrally by one water or methanol molecule and three oxygen atoms from two carboxylate and one phosphonate groups (Supplementary Fig. 6). The ligand exhibits an *exo*-hexadentate coordination fashion, binding to three $Cr_3$ units via two bidentate carboxylate and one phosphonate groups. Adjacent trimetal-phosphocarboxylate clusters are linked by biphenyl backbones of **L** to give a 2D layered structure in the ac plane (Supplementary Fig. 7). Each $Cr_3$ cluster in the plane further coordinates to two carboxylate and one phosphonate groups of three **L** and connect adjacent 2D sheets to form a 3D structure. There are two types of irregular 1D channels with maximal sizes of ~1.0 × 1.2 and ~0.8 × 1.0 nm² that are uniformly lined with $Cr_3$ clusters and hydrophobic *tert*-butyl groups of the **L** ligands. With respect to the topology, **1-Cr** has one vertex, represented by the $Cr_3(CO_2)_4(O_2PO_2)_2$ unit and one bent edge (linker) leading to a 3,6-c net with the $(4^26)_2(4^46^28^9)$ topology.

Single-crystal X-ray crystallography revealed that **1-Mn/Ga/Zr/Ti** are isostructural to **1-Cr** and each has an almost identical 3D structure. The single-crystal X-ray diffraction data for other MOFs were extremely weak, but powder X-ray diffraction (PXRD) study established that all of them are isostructural to **1-Cr** (Fig. 2a, b). PLATON calculations show that the frameworks

of **1**-**Mn/Cr/Ga/Zr** have about 50% of void spaces available for guest inclusion[35]. Circular dichroism (CD) spectra of them made from *R* and *S* enantiomers of H₃**L** are mirror images of each other, suggesting their enantiomeric nature (Supplementary Fig. 9). The +2, +3, and +4 oxidation states of metal ions in **1**-**M** were confirmed by X-ray photoelectron spectroscopy (Supplementary Figs. 19–21).

We thus demonstrated that up to 16 different metal ions could be assembled with the same linker forming the isostructural MOFs. The formation of such a large family of isostructures may be ascribed to the following reasons. First, the SBU formed [M₃(CO₂)₂(O₂PO₂)₂(solvent)₂] in these structures features one metal in the center with octahedral geometry, and two metals in both ends with tetrahedral geometries. It can be seen that all the 16 metals with octahedral geometries are very common in coordination chemistry. It is worth mentioning that Zr⁴⁺ and Ti⁴⁺ ions, etc, tend to have higher coordination numbers. However, the *tert*-butyl groups from the linkers, which are very close to the SBUs, shield the metal ions, and thus make these metals with tetrahedral geometries possible. Second, the acetic acid (HOAc) in MOF synthesis coordinates to the metals to compensate the charges in several cases, thus metals with different charges could fit into this framework. Third, the flexibility of the linkers also provides some freedom to fit metals with different sizes into the same structure. Although ten thousands of MOFs have been reported, there are no reports on specific direct incorporation of such a large number of metals into one isoreticular network, which should greatly enrich its functionalities[4,6,36]. Prior to this report, isostructural MOFs with up to seven different metals have been reported in, for example, MOF-74 [M₂(2,5-dihydroxyter-ephthalate)] type frameworks with divalent metal ions Mg²⁺, Cu²⁺, Zn²⁺, Co²⁺, Mn²⁺, Ni²⁺, and Fe²⁺ [37]. To the best of our knowledge, the present MOF series is with largest number of different metals ever reported. We believe this strategy could be extended to the broad scope of combinations of various metal ions with ligands by carefully designing the geometries and steric shield groups of the organic linkers.

**Thermal and chemical stability**. Similar to other phosphonate-MOFs, **1**-**M** display excellent thermal and chemical stability. Thermogravimetric analysis (TGA) revealed that the guest molecules could be removed in the temperature range from 70 to 180 °C, variable temperature PXRD measurements show that the frameworks of **1**-**Cr/Zr/Mg/Mn** are thermally stable up to 375 °C. (Supplementary Figs. 10 and 13). In contrast, most carboxylate-based MOFs can only withstand temperatures of about 150–350 °C[38,39]. PXRD patterns of **1**-**Mg/Cr/Mn/Zr** indicated that their framework and crystallinity remain intact upon removal of guest molecules. After activating their samples by heating in vacuum (10⁻⁴ Torr) at 100 °C for 6 h, they all exhibited a Type-I sorption behavior, with Brunauer-Emmett-Teller (BET) surface areas of 1110, 1188, 1025 and 1064 m² g⁻¹, respectively, which were calculated from N₂ adsorption isotherms measured at 77 K. After dispersing their crystals in water (23 and 100 °C), aqueous HCl (pH = 4) and NaOH solutions (pH = 9) for 1 week, the frameworks retained their original crystalline structures, as indicated by the almost unchanged intensities and positions of the peaks in PXRD patterns (Fig. 2c and d). The BET surface areas were 1000–1067, 1032–1130, and 1030–1047 m² g⁻¹ for the samples treated for 1 week in boiling water, weak acid and base, respectively (Fig. 3; Supplementary Figs. 14–18); these values are close to that of the pristine MOFs. The relatively high thermal and chemical stability of **1**-**M** compared with most of other carboxylate-based MOFs suggests that −PO₄ groups could promote the tolerance of hybrid platforms toward acidic and alkaline media[29–33].

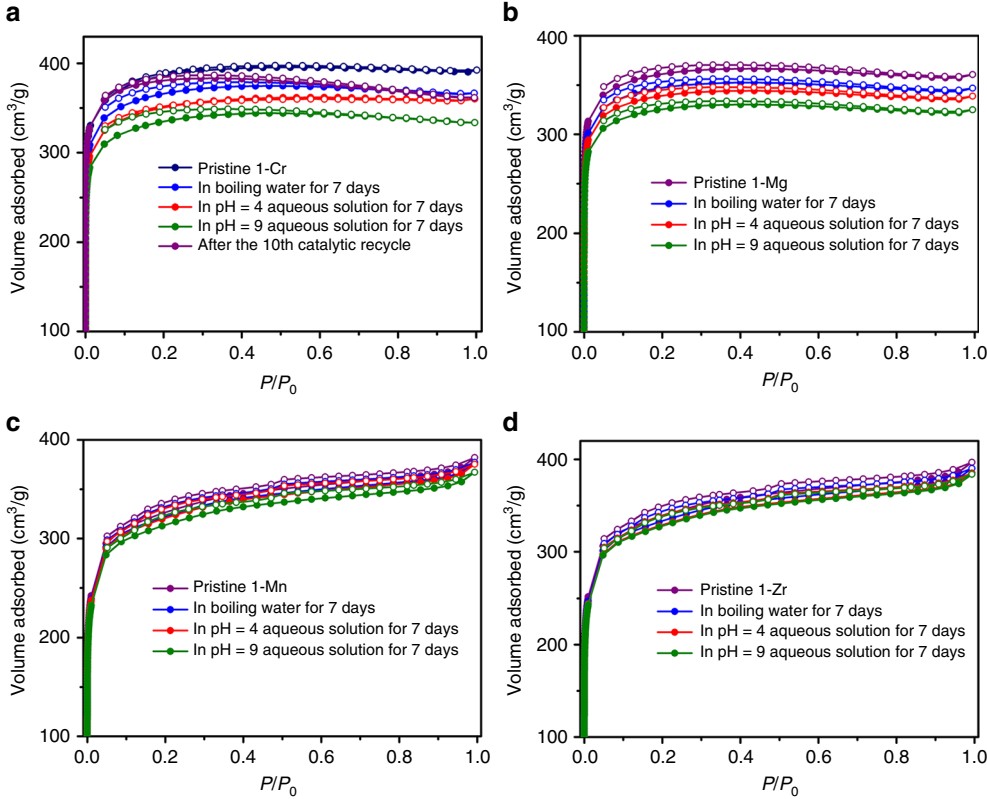

**Fig. 3** N₂ sorption isotherms. N₂ adsorption isotherms (filled symbols) and desorption isotherms (open symbols) at 77 K of MOFs **1**-**Cr** (**a**), **1**-**Mg** (**b**), **1**-**Mn** (**c**), and **1**-**Zr** (**d**) after treatment under different conditions

**Table 1 Allyboration of aldehydes catalyzed by 1-Cr[a]**

| Entry | R | Catalyst | Yield (%)[b] | ee (%)[c] |
|---|---|---|---|---|
| 1 | Ph | **1-Cr** | 98 | 83 (S) |
| 2[d] | Ph | Homo | 81 | 0 |
| 3 | 4-NO$_2$Ph | **1-Cr** | 98 | 99 (S) |
| 4[d] | 4-NO$_2$Ph | Homo | 83 | 0 |
| 5 | 4-ClPh | **1-Cr** | 98 | 92 (S) |
| 6[d] | 4-ClPh | Homo | 81 | 0 |
| 7 | 4-MePh | **1-Cr** | 95 | 84 (S) |
| 8 | 2-MePh | **1-Cr** | 96 | 86 (S) |
| 9 | 2-NO$_2$Ph | **1-Cr** | 98 | 98 (S) |
| 10 | 2-Thienyl | **1-Cr** | 98 | 88 (R)[e] |
| 11 | DBBA | **1-Cr** | 21 | 0 |
| 12[d] | DBBA | Homo | 93 | 0 |

DBBA: 3,5-dibenzyloxybenzaldehyde
[a] Reaction conditions: **2** (0.10 mmol), **3** (0.12 mmol), and (S)-**1-Cr** (2 mol% catalyst based on MOF) in DCE (0.8 mL), −10 °C, 12 h
[b] Isolated yield
[c] Determined by chiral HPLC analysis (letters in brackets specify the excess isomer)
[d] A 1:1 mixture of Cr(acac)$_3$ and (S)-Me$_2$**L** (6 mol% catalyst based on Cr) was used as the catalyst
[e] The R enantiomer was produced using (R)-**1-Cr** as the catalyst

**Table 2 Propargylation of aldehydes catalyzed by 1-Cr[a]**

| Entry | R | Catalyst | Yield (%)[b] | ee (%)[c] |
|---|---|---|---|---|
| 1 | Ph | **1-Cr** | 92 | 92 (R) |
| 2[d] | Ph | Homo | 81 | 0 |
| 3 | 4-MeOPh | **1-Cr** | 96 | 96 (R) |
| 4[d] | 4-MeOPh | Homo | 78 | 0 |
| 5 | 4-ClPh | **1-Cr** | 93 | 92 (R) |
| 6 | 4-BrPh | **1-Cr** | 91 | 92 (R) |
| 7 | 4-NO$_2$Ph | **1-Cr** | 94 | 91 (R) |
| 8 | 4-COOMePh | **1-Cr** | 92 | 94 (R) |
| 9 | 3-MeOPh | **1-Cr** | 90 | 99 (R) |
| 10 | 1-Naphthyl | **1-Cr** | 92 | 93 (R) |
| 11 | PhCH=CH | **1-Cr** | 90 | 97 (R) |
| 12 | DBBA | **1-Cr** | 0 | 0 |
| 13[d] | DBBA | Homo | 90 | 0 |

DBBA: 3,5-dibenzyloxybenzaldehyde
[a] Reaction conditions: **2** (0.10 mmol), **5** (0.15 mmol), and (S)-**1-Cr** (2 mol% catalyst based on MOF) in DCE (0.8 mL), −10 °C, 12 h
[b] Isolated yield
[c] Determined by chiral HPLC analysis (letters in brackets specify the excess isomer)
[d] A 1:1 mixture of Cr(acac)$_3$ and (S)-Me$_2$**L** (6 mol% catalyst based on Cr) was used as the catalyst

**Heterogeneous asymmetric catalysis**. With the MOFs in hand, we set out to study the catalytic activity of coordinatively unsaturated metal centers located in the channel surfaces. Classic reactions including allylboration, propargylation, Friedel–Crafts alkylation, and sulfoxidation that are relevant to the synthesis of pharmaceuticals were examined with the MOFs through the judicious choice of metal ions[40–42]. Chiral metal phosphates have been employed as homogeneous catalysts for asymmetric organic reactions, but not for the present four reactions[34,43,44]. After screening **1-M** under different reaction conditions, **1-Cr** was found to be the best catalyst in terms of both conversion and enantioselectivity in conjugate additions of allylboronic acid esters to aromatic aldehydes[45]. Typically, 2 mol% loadings of **1-Cr** catalyze the reaction of allylboronic acid pinanediol ester with benzaldehyde in 1,2-dichloroethane (DCE) at −10 °C to give the desired product in 98% isolated yield with 83% ee after 12 h (Table 1). Both electron-withdrawing and electron-donating groups on the aromatic ring were tolerated in the reaction, affording the products in 95–98% yields with 84–98% ee. Heterocyclic compound 2-thienylaldehyde was also tolerated, giving the product in 98% yield with 88% ee.

Inspired by the allylboration results, we subsequently studied the propargylation of aldehydes, which is more challenging owing to the lower reactivity of the allenyl boronic acid pinacol ester relative to pinacol allylboronate[40,46]. 2 mol% loading of **1-Cr** promoted propargylation of aldehydes for both aromatic aldehydes with electron-donating and electron-withdrawing groups on the aromatic rings, furnishing the homopropargylic alcohols in 90–96% isolated yields with 91–99% ee after 12 h (Table 2). When 1-naphthaldehyde was used as a substrate, 92% yield and 93% ee were obtained. The aliphatic aldehyde PhCH=CHCHO was converted to the products in 90% yield with 97% ee.

**1-Mg** was the excellent catalyst for the asymmetric Friedel–Crafts alkylation reaction of nitroalkenes with pyrroles[41,47]. Specially, 2 mol% loading of **1-Mg** catalyzed the reaction of trans-β-nitrostyrene with pyrrole producing the alkylation product in 89% isolated yields with 90% ee after 12 h (Table 3). A variety of aryl-substituted nitroalkenes, containing electro-donating and electro-withdrawing groups on the aromatic rings was tolerated, giving rise to 82–90% isolated yields with 88–99% ee. The highest enantioselectivity was attained with 1-(4-bromophenyl) nitroethylene, affording 90% yield and 99% ee.

**1-Mn** is highly active and selective in catalytic oxidation of sulfides with aqueous 30% H$_2$O$_2$ (1.2 equiv.) as the oxidant[42,48]. Typically, at 2 mol% loading of (R)-**1-Mn** in DCE at −30 °C, the oxidation of phenyl methyl sulfide afforded the (R)-sulfoxide in 93% isolated yield with 92% ee after 4 h (Table 4). Increasing the oxidant loading to 1.5 or 2.0 equiv could increase the reaction rate but not enantioselectivity. A variety of substrates with different substituents on the aromatic rings were tolerated, and in all cases, moderate to good enantioselectivities were obtained, which ranged from 55 to 93% for those with electron-withdrawing or electron-donating groups. Remarkably, all of the above reactions proceeded without over-oxidation of sulfides to sulfones, as revealed by [1]H NMR analysis. To our knowledge, the stereoselectivity and chemoselectivity of **1-Mn** exceeded those of reported heterogeneous catalysts and were comparable even to those of the best metal-based homogeneous systems[42,49], suggesting its utility in other asymmetric oxidations[50].

To exclude coexistence of free phosphoric acid and Lewis acid in the MOF catalysts, **1-Cr/Mg/Mn** was used to catalyze hydrogenation of quinoxalines with Hanztsch esters, which can be catalyzed by Brønsted acids and cannot by Lewis acids (Supplementary Table 6)[51]. It was found that, under identical reaction conditions, **1-Cr/Mg/Mn** cannot provide any product even at elevated temperature, but both (S)-Me$_2$**L** and (S)-H$_3$**L** containing free phosphoric acids provided the product in good to high yields. The result suggests there does not exist free phosphoric acid in the MOF. Furthermore, after completion of the reactions catalyzed by **1-Cr/Mg/Mn**, the supernatant was condensed and [31]P NMR showed that there was no free

**Table 3 Friedel–Crafts alkylation of pyrrole with nitroalkenes catalyzed by 1-Mg[a]**

| Entry | R | Catalyst | Yield (%)[b] | ee (%)[c] |
|---|---|---|---|---|
| 1 | Ph | **1-Mg** | 89 | 90 (S) |
| 2[d] | Ph | Homo | 71 | 0 |
| 3 | 4-MeOPh | **1-Mg** | 90 | 96 (S) |
| 4[d] | 4-MeOPh | Homo | 68 | 0 |
| 5 | 4-EtOPh | **1-Mg** | 90 | 94 (S) |
| 6[d] | 4-EtOPh | Homo | 75 | 0 |
| 7 | 4-BrPh | **1-Mg** | 90 | 99 (S) |
| 8 | 4-COOMePh | **1-Mg** | 84 | 88 (S) |
| 9 | 2-MeOPh | **1-Mg** | 87 | 88 (S) |
| 10 | 3-ClPh | **1-Mg** | 82 | 91 (S) |

[a] Reaction conditions: **7** (0.3 mmol), **8** (0.1 mmol), and (S)-**1-Mg** (2 mol % catalyst based on MOF) in DCE (0.8 mL), −10 °C, 12 h
[b] Isolated yields
[c] Determined by chiral HPLC analysis (letters in brackets specify the excess isomer)
[d] A 1:1 mixture of Mg(NO$_3$)$_2$·6H$_2$O and (S)-Me$_2$**L** (6 mol% catalyst based on Mg) was used as the catalyst

**Table 4 Oxidation of sulfide catalyzed by 1-Mn[a]**

| Entry | R | Catalyst | Conv. (%)[b] | Yield (%)[c] | ee (%)[d] |
|---|---|---|---|---|---|
| 1 | Ph | **1-Mn** | 99 | 93 | 92 (S) |
| 2[e] | Ph | Homo | 71 | 67 | 0 |
| 3 | 4-MePh | **1-Mn** | 99 | 93 | 92 (S) |
| 4[e] | 4-MePh | Homo | 63 | 59 | 0 |
| 5 | 4-MeOPh | **1-Mn** | 95 | 92 | 84 (S) |
| 6[e] | 4-MeOPh | Homo | 80 | 75 | 0 |
| 7 | 4-ClPh | **1-Mn** | 96 | 94 | 89 (S) |
| 8 | 3-FPh | **1-Mn** | 98 | 91 | 87 (S) |

[a] Reaction conditions: **10** (0.1 mmol), (S)-**1-Mn** (2 mol % catalyst based on MOF), 1.2 equiv. aqueous H$_2$O$_2$ in DCE (0.8 mL), −30 °C, 4 h. No over-oxidized sulfone byproducts were detected
[b] Determined by $^1$H NMR analysis
[c] Isolated yield
[d] Determined by chiral HPLC analysis (letters in brackets specify the excess isomer)
[e] A 1:1 mixture of MnCl$_2$·4H$_2$O and (S)-Me$_2$**L** (6 mol% catalyst based on Mn) was used as the catalyst

phosphoric acid leaked from the MOF in the solution, which indicated that the reaction was not catalyzed by a free phosphoric acid (Supplementary Figs. 1–4). Besides, in the FT-IR spectra, the absence of any stretching vibration bands around 2350 cm$^{-1}$ also implies that there are no free P−OH groups in **1-M** (Supplementary Fig. 11)[52], consistent with the single-crystal X-ray diffraction. Taken together, the experimental results strongly suggested that the terminal metal centers, in which the terminally coordinated guests can be readily removed by heating under vacuum, in the trimeric units of the MOF structure were catalytically active sites responsible for the examined asymmetric reactions.

To evaluate the contribution of pore structures of the MOFs to the above asymmetric catalysis, we examined the catalytic activities of molecular metal phosphonates. Metal phosphonates crystallized from a 1:1 mixture of (S)-Me$_2$**L** and related metal salts at r.t. can be formulated as [Mg$_3$(Me$_2$**L**)$_6$(H$_2$O)$_8$]·4H$_2$O, [Mn$_3$(Me$_2$**L**)$_6$(H$_2$O)$_8$]·3H$_2$O and [Cr$_3$(Me$_2$**L**)$_6$(H$_2$O)$_5$][OH]$_3$·3H$_2$O on the basis of elemental analysis and spectroscopic techniques. Single-crystal-X-ray diffraction revealed that the three metals were in a bent arrangement and adjacent metals were linked by one bidentate phosphonate groups. The central metal was coordinated by six oxygen atoms from two phosphonate groups and four water molecules, and each terminal metal was coordinated by five oxygen atoms from three phosphonate groups and two water molecules (Supplementary Fig. 8). Mixtures of (S)-H$_3$**L** and metal salts are not soluble on common organic solvents, preventing studying their homogeneous catalytic activities. Interestingly, metal phosphonates crystallized from a 2:3 mixture of (S)-H$_3$**L** and the corresponding metal salts at r.t. are isostructural to **1-Mg/Mn/Cr** (Supplementary Table 1). Under otherwise identical conditions, control reactions with (S)-Me$_2$**L**, (S)-H$_3$**L**, and the trimetallic (S)-Me$_2$**L-Mg/Mn/Cr** afforded 0% ee and moderate to good yields (47–88%) in all cases (Tables 1–4 and Supplementary Tables 2–5). It should be noted that chiral BINOL-derived phosphoric acids and/or their metal salts are reported to be efficient catalysts for asymmetric allylboration, propargylation, Friedel–Crafts alkylation, and sulfoxidation[45–48]. The bulky substituents such as 9-anthryl and

2,4,6-triisopropylphenyl groups at 3,3′-positons with the BINOL backbone could increase steric interactions between substrates and catalysts and induce high stereoselectivity. In contrast, the phosphoric acids Me$_2$**L** and H$_3$**L** have a slightly flexible biphenol backbone with less bulky *tert*-butyl groups at 3,3′-positions, which may be responsible for their inability of asymmetric induction in the tested reactions. Therefore, it may be reasonable to conclude that, in the present MOFs, chiral phosphono-carboxylate groups of 1,1′-biphenol, together with the metal ions and phenyl rings create a porous microenvironment, which is believed to be responsible for the high degrees of catalytic activity and selectivity, by concentrating reactants and providing additional steric and electronic effects around the Lewis acid metal sites.

Multiple experiments were performed to prove that the MOF catalysts are true heterogeneous and reusable catalysts. Upon completion of alkylation of 4-methoxy-benzaldehyde, **1-Cr** could be recovered in quantitative yield and used repeatedly without the deterioration of activity and enantioselectivity for the following ten runs (86–96% isolated yields and 93–96% ee for 1–10 run). After 10 cycles, both the PXRD pattern and BET surface area of the recovered **1-Cr** remained almost the same as those of the pristine sample (Figs. 2c and 3a). The supernatant from the alkylation after filtration through a regular filter did not afford any additional alkylation product. ICP-OES (Inductively coupled plasma optical emission spectrometry) analysis of the product solution indicated almost no loss of the Cr ion (<0.002%) from the structure per cycle. Similar results were found for **1-Mg** and **1-Mn** in alkylation of nitroalkenes with pyrroles and oxidation of sulfides, respectively (Supplementary Figs. 12 and 18). The sterically more demanding substrate 3,5-dibenzyloxybenzalde-hyde could not be a suitable substrate for allylboration or propargylation reactions, and only a very small amount of product (21% and <5% conversion, background reaction existed for allylboration) was detected in each case (Table 1, entry 11 and Table 2, entry 12). Meanwhile, the corresponding homogeneous catalyst chromium phosphonates (a 1:1 mixture of Cr(acac)$_3$ and Me$_2$**L**, 6 mol% loading catalyst based on Cr) still afforded the product in 93 and 90% yield, respectively, indicating that the bulky substrate could not diffuse into the MOF catalyst efficiently

**a**

**b**

**Fig. 4** Schematic representation of production elaboration. **a** The synthesis of 3,4-allenol and **b** pyrrolo[3,2-c]pyridine

and the heterogeneous catalysis does not occur mainly at surface sites. This point was also supported by the fact that ground and unground particles of **1**-**Cr** exhibited similar catalytic performance (95 vs. 96% conversions in 12 h) in catalyzing propargylation of 4-methoxy-benzaldehyde.

**Production elaboration**. As key building blocks of bioactive compounds and natural products, pyrrolopyridine and allenol derivatives have inspired much interest in developing methodologies for their catalytic asymmetric synthesis[50,53], but their synthesis has not been achieved with a heterogeneous catalyst[3,54]. The synthetic utility of our catalytic systems was exemplified in the construction of two derivatives of pyrrolo[3,2-c]pyridine and 3,4-allenol (Fig. 4). (R)-**6a** was obtained in 90% yield with 92% ee from a gram-scale synthesis using propargylation of benzaldehyde catalyzed by 2 mol% (catalyst based on Mn) **1**-**Cr** and was converted to **12** via CuBr-mediated Crabbe homologation reaction in 91% yield with 92% ee. Starting from nitroalkene and pyrrole, the use of 6 mol % (S)-**1**-**Mg** afforded the indole derivative (R)-**9b** in 90% yield (1.2 g) with 95% ee. Reduction of the nitro group of **9b** and then Pictet–Spengler cyclization with aldehydes gave (R)-**13** in 90% yield with 94 % ee.

## Discussion

In conclusion, we have developed a broad library of 16 isostructural MOFs featuring excellent stability, permanent porosity, and homochirality based on an enantiopure phosphocarboxylate ligand of 1,1′-biphenol with pendant *tert*-butyl group at the 3,3′-position. The use of phosphonates as linkers for the construction of MOFs provides strong metal–oxygen coordination bonds to stabilize the frameworks and chiral Lewis acid metal acids to catalyze organic transformations, whereas appending bulky *tert*-butyl groups provides a way to protect hydrolytically susceptible coordination backbones through kinetic blocking and to exert steric and electronic control over catalytic reactions simultaneously. The series of MOF materials shows a regular variation of the Lewis acidity, which can be readily tuned by incorporating diverse metal ions, and has been demonstrated to catalyze four types of asymmetric organic transformations with activity, enantioselectivity, recyclability, and environmental benignity—a set of characteristics that has remained challenging to engineer together in heterogeneous catalysis. In contrast, the corresponding homogeneous systems cannot enantioselectively catalyze the

test reactions. Our synthetic strategy thereby provides an approach to achieve single-sited heterogeneous catalysts that combine catalytic activity and versatility, enantioselectivity, and recyclability for a broad scope of asymmetric organic reactions and may find applications in practical synthesis of pharmaceutical and fine chemicals.

## Methods

**Synthesis of MOFs 1-Mg/Ca/Mn/Co/Ni/Cu/Zn/Pd/Pb**. A mixture of Mg $(NO_3)_2 \cdot 6H_2O$ $(Ca(NO_3)_2 \cdot 4H_2O/MnCl_2 \cdot 6H_2O/Co(NO_3)_2 \cdot 6H_2O/Ni(NO_3)_2 \cdot 6H_2O/Cu(NO_3)_2 \cdot 6H_2O/Zn(NO_3)_2 \cdot 6H_2O/PdCl_2/Pb(NO_3)_2 \cdot 6H_2O$, 0.04 mmol), $H_3L$ (20 mg, 0.03 mmol), MeOH (5 mL), and HOAc (0.5 mL) was sealed in a 10 mL vial with a screw cap and heated at 80 °C for 1 day. The mixture was cooled to room temperature, then block-like crystals were obtained, washed with ether and dried in air. Yield: 80, 75, 72, 76, 72, 68, 77, 82, and 61% for **1-Mg/Ca/Mn/Co/Ni/Cu/Zn/Pd/Pb** (based on metal salts), respectively.

**Synthesis of MOFs 1-Cr/Fe/Al/Ga**. A mixture of $Cr(acac)_3$ $(Fe(acac)_3/AlCl_3/GaCl_3$, 0.04 mmol), $H_3L$ (20 mg, 0.03 mmol), MeOH (1 mL), and HOAc (0.5 mL) was sealed in a 10 mL vial with a screw cap and heated at 80 °C for 2 days. The mixture was cooled to room temperature, then block-like crystals were obtained, washed with ether, and dried in air. Yield: 80, 68, 60, and 80% (based on metal salts) for **1-Cr/Fe/Al/Ga**, respectively.

**Synthesis of MOFs 1-Zr/Ti/Sn**. A mixture of $Zr(NO_3)_4 \cdot 5H_2O/(Ti(O-iPr)_4/SnCl_4 \cdot 6H_2O$, 0.04 mmol), $H_3L$ (20 mg, 0.03 mmol), MeOH (3 mL), and HOAc (0.5 mL) was sealed in a 10 mL vial with a screw cap and heated at 80 °C for 2 days. The mixture was cooled to room temperature, then yellow block-like crystals were obtained, washed with ether, and dried in air. Yield: 70, 70, and 52% for **1-Zr/Ti/Sn** (based on metal salts), respectively.

**Characterization**. The solid state CD spectra were conducted on a J-800 spectropolarimeter (Jasco, Japan). $N_2$ sorption measurements were performed with Micrometritics ASAP 2020. Physisorption. EA were conducted with an EA1110 CHNS-0 CE elemental analyzer. The IR spectra were collected (400–4000 cm$^{-1}$ region) on a Nicolet Magna 750 FT-IR spectrometer. TGA experiments were conducted with STA449C integration thermal analyzer. ICP-OES data were performed on 7300DV ICP-OES. PXRD were collected on a DMAX2500 diffractometer equipped with a Cu tube. NMR data were carried out on a MERCURYplus 400 MHz NMR spectrometer. ES–MS were recorded on a Finnigan LCQ mass spectrometer. HPLC were performed on a YL-9100 HPLC and Shimadzu 2010A. See Supplementary Methods for the procedures and characterization data of compounds not listed in this part.

**Single-crystal X-ray diffraction**. Single-crystal XRD data for the five MOFs and related metal phosphonates were collected on a Bruker Smart APEX II CCD diffractometer and a Bruker D8 VENTURE CMOS photon 100 diffractometer with helios mx multilayer monochromator Cu–Kα radiation ($\lambda = 1.54178$ Å) at 123 and 173 K. The empirical absorption correction was applied by using the SADABS program[55]. The structure was solved using direct method, and refined by full-

matrix least-squares on $F2^{56,57}$. All non-hydrogen atoms are refined aniso-
tropically, except the guest molecules. Owing to the weak diffraction, none of guest
molecules could be found in difference Fourier maps and all the phenyl rings are
constrained to ideal six-membered rings. Contributions to scattering due to these
highly disordered solvent molecules were removed using the SQUEEZE routine of
PLATON for MOFs; structures were then refined again using the data generated
under OLEX2-1.2. Furthermore, it was necessary to use constraints to control the
geometry of the aromatic rings and restraints to enforce chemically sensible bond
lengths and angles in the *tert*-butyl groups. Crystal data and details of the data
collection are given in Supplementary Tables 1, 7–9, and 10–16.

**General procedure for asymmetric reactions catalyzed by 1-Cr/Mg/Mn.**
Allylboration of aldehydes: To a 10 mL flame-dried Schlenk tube, the activated **1-
Cr** (2 mol%, catalyst based on MOF), aldehydes (0.1 mmol), and dry DCE (0.5 mL)
were added. The reaction mixture was stirred for 1 h and then cooled to −10 °C
followed by dropwise addition of dry DCE (0.3 mL)-containing allylboronic acid
pinacol ester (0.12 mmol). The mixture was stirred for 12 h at this temperature and
then purified by flash chromatography using EtOAc and petroleum ether. For the
possible catalytic cycle, see Supplementary Fig. 5.

Propargylation of aldehydes: To a 10 mL flame-dried Schlenk tube, the activated
**1-Cr** (2 mol%, catalyst based on MOF), aldehydes (0.1 mmol), and dry DCE
(0.5 mL) were added. The reaction mixture was stirred for about 1 h and then
cooled to −10 °C followed by dropwise addition of dry DCE (0.3 mL)-containing
allenylboronic acid pinacol ester (0.15 mmol). The mixture was stirred for 12 h at
this temperature and then purified by flash chromatography using EtOAc and
petroleum ether.

Friedel–Crafts alkylation: To a 10 mL flame-dried Schlenk tube, the activated **1-
Mg** (2 mol%, catalyst based on MOF), pyrrole (0.3 mmol), and dry DCE (0.3 mL)
were added. The reaction mixture was stirred for 1 h and then cooled to −10 °C
followed by dropwise addition of dry DCE (0.5 mL)-containing nitroalkenes
(0.1 mmol). The mixture was stirred for 12 h at this temperature and then purified
by flash chromatography using EtOAc and petroleum ether.

Oxidation of sulfides: A 10 mL flame-dried Schlenk tube equipped with a stir
bar and the activated **1-Mn** (2 mol%, catalyst based on MOF) was added followed
by addition of sulfides (0.1 mmol). The mixture was added with DCE (0.5 mL) and
stirred for 1 h, then cooled to −30 °C. Aqueous $H_2O_2$ (30%, 14 μL, 1.2 equiv.) in
DCE (0.3 mL) was added dropwise to the suspension and stirred for 4 h at this
temperature. Purification by column chromatography on silica gel using EtOAc
and petroleum ether as an eluent gave the desired sulfoxide.

Full experimental details and characterization of intermediates and Ligand are
given in Supplementary Information, see Supplementary Figs. 57–62. For NMR
and HPLC spectra of the products obtained by catalysis in this article, see
Supplementary Figs. 22–56 and 63–111.

**Data availability**. The X-ray crystallographic coordinates for the structures
reported in this article have been deposited at the Cambridge Crystallographic Data
Centre (CCDC), under deposition numbers CCDC 1477932-1477936 and
1576522-1576523. These data can be obtained free of charge from The Cambridge
Crystallographic Data Centre via www.ccdc.cam.ac.uk/data_request/cif. All other
data supporting the findings of this study are available within the article and
its Supplementary Information, or from the corresponding author upon reasonable
request.

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

# ARTICLE

37. Yaghi, et al. Rod packings and metal–organic frameworks constructed from rod-shaped secondary building units. *J. Am. Chem. Soc.* **127**, 1504–1508 (2005).

38. Mondloch, J. E. et al. Vapor-phase metalation by atomic layer deposition in a metal–organic framework. *J. Am. Chem. Soc.* **135**, 10294–10297 (2013).

39. Kang, I. J., Khan, N. A., Haque, E. & Jhung, S. H. Chemical and thermal stability of isotypic metal–organic frameworks: effect of metal ions. *Chem. Eur. J.* **17**, 6437–6442 (2011).

40. Ding, C. & Hou, X.-L. Catalytic asymmetric propargylation. *Chem. Rev.* **111**, 1914–1937 (2011).

41. Poulsen, T. B. & Jørgensen, K. A. Catalytic asymmetric Friedel–Crafts alkylation reactions—copper showed the way. *Chem. Rev.* **108**, 2903–2915 (2008).

42. Fujisaki, J., Matsumoto, K., Matsumoto, K. & Katsuki, T. Catalytic asymmetric oxidation of cyclic dithioacetals: highly diastereo- and enantioselective synthesis of the *S*-oxides by a chiral aluminum(salalen) complex. *J. Am. Chem. Soc.* **133**, 56–61 (2011).

43. Hatano, M., Moriyama, K., Maki, T. & Ishihara, K. Which is the actual catalyst: chiral phosphoric acid or chiral calcium phosphate. *Angew Chem. Int. Ed.* **49**, 3911–3914 (2010).

44. Alix, A., Lalli, C., Retailleau, P. & Masson, G. Highly enantioselective electrophilic α-bromination of enecarbamates: chiral phosphoric acid and calcium phosphate salt catalysts. *J. Am. Chem. Soc.* **134**, 10389–16614 (2012).

45. Jain, P. & Antilla, J. C. Chiral Brønsted acid-catalyzed allylboration of aldehydes. *J. Am. Chem. Soc.* **132**, 11884–11886 (2010).

46. Jain, P., Wang, H., Houk, K. N. & Antilla, J. C. Brønsted acid catalyzed asymmetric propargylation of aldehydes. *Angew Chem. Int. Ed.* **124**, 1391–1394 (2012).

47. Sheng, Y. F., Gu, Q., Zhang, A. J. & You, S. L. Chiral Brønsted acid-catalyzed asymmetric Friedel–Crafts alkylation of pyrroles with nitroolefins. *J. Org. Chem.* **74**, 6899–6901 (2009).

48. Liao, S., Coric, I., Wang, Q. & List, B. Activation of $H_2O_2$ by chiral confined Brønsted acids: a highly enantioselective catalytic sulfoxidation. *J. Am. Chem. Soc.* **134**, 10765–10768 (2012).

49. Chen, Y., Tan, R., Zhang, Y., Zhao, G. & Yin, D. Dendritic chiral salen titanium (IV) catalysts enforce the cooperative catalysis of asymmetric sulfoxidation. *ChemCatChem* **7**, 4066–4075 (2015).

50. Hermans, I., Spier, E. S., Turrà, N. & Baiker, A. Selective oxidation catalysis: opportunities and challenges. *Top. Catal.* **52**, 1162–1174 (2009).

51. Rueping, M., Tato, F. & Schoepke, F. R. The first general, efficient and highly enantioselective reduction of quinoxalines and quinoxalinones. *Chem. Eur. J.* **16**, 2688–2691 (2010).

52. Dar, A. A. et al. Dimensionality alteration and intra- versus inter-SBU void encapsulation in zinc phosphate frameworks. *Inorg. Chem.* **55**, 5180–5190 (2016).

53. Shumaila, A. M. A., Puranik, V. G. & Kusurkar, R. S. Synthesis of tetrahydro-5-azaindoles and 5-azaindoles using Pictet–Spengler reaction—appreciable difference in products using different acid catalysts. *Tetrahedron* **67**, 936–942 (2011).

54. Fraile, J. M., Garcia, J. & Mayoral, J. A. Noncovalent immobilization of enantioselective catalysts. *Chem. Rev.* **109**, 360–417 (2009).

55. Sheldrick, G. M. *SADABS, Program for Empirical Absorption Correction of Area Detector Data* (University of Göttingen, Göttingen, Germany, 1996).

56. Sheldrick, G. M. *SHELXTL-97, Program for Crystal Structure Refinement* (University of Göttingen, Göttingen, Germany, 1997).

57. Sheldrick, G. M. *SHELXTL-2014, Program for Crystal Structure Refinement* (University of Göttingen, Göttingen, Germany, 2014).

## Acknowledgements

This work was financially supported by the National Science Foundation of China (Grants 21371119, 21431004, 21401128, 21522104, and 21620102001), the National Key Basic Research Program of China (Grants 2014CB932102 and 2016YFA0203400), Key Project of Basic Research of Shanghai (17JC1403100), and the Shanghai "Eastern Scholar" Program.

## Author contributions

Y.C. and Y.L. conceived and designed the research. Y.P., X.C., X.L., and Y.L. performed the experiments. Y.C., Y.L., Y.P., and X.C. analyzed the data and co-wrote the manuscript.

## Additional information

**Competing interests:** The authors declare no competing financial interests.

9