## [Peer Review File · Nature Communications]

Reviewer #1 (Remarks to the Author):

In this work, Liu and Cui et al. have reported a series of isostructural phosphonate MOFs and their applications in heterogeneous asymmetric catalytic reactions. The enantiomeric excesses are up to 99% in a series of organic reactions, including allylboration of aldehydes, propargylation of aldehydes and Friedel-Crafts alkylation of pyrrole. Considering that it is challenging for the synthesis of chiral MOFs, these catalytic results are encouraging. Acceptance of this work is recommended after the following aspect is clarified.

It is a little hard to understand why the corresponding homogeneous catalysts haven't induced any enantioselectivities at all in these reactions (0% ee in each reaction). As shown in the footnotes of Tables 1-4, the homogeneous catalysts were prepared by a 1:1 mixture of metal salts and (R)-Me₂L. Both the name and structure of (R)-Me₂L are missing in this manuscript. How about the structures of the homogeneous catalysts as prepared in this way? It is unclear what the catalytically active units in the MOFs are responsible for the asymmetrical reactions. I strongly suggest the authors to examine the structures of the homogeneous catalysts used in the comparative experiments before they draw a conclusion about the confinement effect of MOFs on the enantioselectivities.

There are several minor suggestions. Variable temperature PXRD experiments should be also carried out to examine the thermal stabilities of the MOFs. It is said that the catalyst amount is "4 mol% catalyst". Does it mean that 4 mol metal %? In the contrast test, the homogeneous catalyst amount is 5 mol%. Considering that the catalyst amounts are not the same, the results may be inappropriate for the comparison of the catalytic activities of different catalysts.

Reviewer #2 (Remarks to the Author):

Although intensive progresses have been achieved in the field of heterogeneous asymmetric catalysis by MOFs, the tune of steric and electronic effects while retaining isorecticular structures is challenging, Cui and coworkers developed an attractive strategy to construct isorecticular MOFs using chiral phosphoric acid containing ligand and different kinds of metal ions, and disclosed their excellent performances in the heterogeneous asymmetric catalyses, which might fit the criteria of publication in *Nat. Commun.*, if the authors could properly rationalize the following aspects.

- 1) It is interesting to tolerate several kinds of metal ions with different coordination abilities in MOFs with isorecticular structures, what is the critical factor inside? What is the state-of-the-art of this point? And is it possible to extend this strategy to the broader scope of combinations of other ligands and metal ions?
- 2) In Fig. 3a, some curves like "Pristine 1-Cr" showed a special dependence of adsorption volume to P/P₀, and higher adsorption was achieved under a medium value of P/P₀. Why? Is there any pressure response of the MOF structure?
- 3) With respect of asymmetric allylboration of aldehyde catalyzed by 1-Cr, there might be different rationalizations of asymmetric inductions. Cause that the chiral phosphoric acid also could catalyze this reaction and give high ee value, see *J. Am. Chem. Soc.* 2012, 134, 2716–2722; of course, the combination of BINOL and Lewis acidic metal ion like Sn also could lead to asymmetric induction, see *J. Am. Chem. Soc.* 2008, 130, 8481–8490. Thus, there might be at least two different modes: the first is the residual free phosphoric acid moieties without coordination to chromiums that might be responsible for the asymmetric induction, and the second possibility will be like the statement by the authors-induced by the Lewis acidity of the chromium centers with chiral environments around them. The question is how to distinguish those two kinds of possibilities? And which one will be responsible for the good performance of the title catalysis in paper? And how does it work? If the second mode is more possible, which one among the three chromiums in a unit will be more likely to take charge of chiral induction?

Response to Reviewer's Comments

Reviewer #1 (Remarks to the Author):

Comments:

In this work, Liu and Cui et al. have reported a series of isostructural phosphonate MOFs and their applications in heterogeneous asymmetric catalytic reactions. The enantiomeric excesses are up to 99% in a series of organic reactions, including allylboration of aldehydes, propargylation of aldehydes and Friedel-Crafts alkylation of pyrrole. Considering that it is challenging for the synthesis of chiral MOFs, these catalytic results are encouraging. Acceptance of this work is recommended after the following aspect is clarified.

(a) It is a little hard to understand why the corresponding homogeneous catalysts haven't induced any enantioselectivities at all in these reactions (0% ee in each reaction).

[Author reply]. Thanks for your kind and great comment.

1) To further confirm the corresponding homogeneous catalysts cannot induce enantioselectivities, more control experiments were performed. Under otherwise identical conditions, all control reactions with (*S*)-Me₂L, (*S*)-H₃L, and metal phosphonates [(*S*)-Me₂L-M] afforded **0% ee** and moderate to good yields (47-88%). Notably, metal phosphonates prepared from a 2:3 mixture of (*S*)-H₃L and metal salts are not soluble in common organic solvents.

Please see **Lines 26-30/Page 10/MS** and **Tables S2-S5/Page S13-S16/SI**.

2) It should be noted that chiral BINOL-derived phosphoric acids and/or their metal salts are reported to be efficient catalysts for asymmetric allylboration, propargylation, Friedel-Crafts alkylation and sulfoxidation. The bulky substituents such as 9-anthryl and 2,4,6-triisopropylphenyl groups at 3, 3'-positions with the BINOL backbone could increase steric interactions between substrates and catalysts and induce high stereoselectivity. [e.g. **Allylboration of aldehydes**: *J. Am. Chem. Soc.*, **132**, 11884 (2010); **Propargylation of aldehydes**: *Angew. Chem., Int. Ed.*, **124**, 1420 (2012); **Friedel-Crafts alkylation**: *J. Org. Chem.*, **74**, 6899 (2009); **Oxidation of sulfide**: *J. Am. Chem. Soc.*, **134**, 10765 (2012)].

In contrast, the phosphoric acids Me₂L and H₃L have a slightly flexible biphenol backbone with less bulky *tert*-butyl groups at 3,3'-positions, which may be responsible for their inability of asymmetric induction in the tested reactions.

This point was mentioned in the main text. Please see **Lines 30-36/Page 10/MS**.

(b) As shown in the footnotes of Tables 1-4, the homogeneous catalysts were prepared by a 1:1 mixture of metal salts and (*R*)-Me₂L. Both the name and structure of (*R*)-Me₂L are missing in this manuscript. How about the structures of the homogeneous catalysts as prepared in this way?

[Author reply].

1) We added the name and structure of (*S*)-Me₂L in the SI.

Please see **Section 2.1 (yellow part)/Pages S2-S3 & Page S124/SI**.

2) Fortunately, we have got the **single-crystals of metal phosphonate of (*S*)-Me₂L (Me₂L-Cr/Mg/Mn)** by slow evaporation of their solutions. Single-crystal X-ray diffraction showed that both Me₂L-Mn and Me₂L-Mg have a similar trimetal structure. The diffraction data for Me₂L-Cr had extremely weak diffraction, but cell parameter determination showed that it has a isostructural structure. Based on element Analysis, ICP-MS (P:M ratios), FT-IR and X-ray diffraction, they can be formulated as [Cr₃(Me₂L)₆(H₂O)₅][OH]₃·3H₂O, [Mg₃(Me₂L)₆(H₂O)₈]·4H₂O and [Mn₃(Me₂L)₆(H₂O)₈]·3H₂O.

Please see **Lines 19-25/Page 10/MS** and **Section 2.2d/Pages S6-S8/SI**.

The structure of Me₂L-Mn/Mg: The complex crystallizes in the chiral space group *P*2₁2₁2₁. The structure has a trimeric M unit, which is linked together by two bridging phosphonate groups of six Me₂L. The central metal is coordinated by six oxygen atoms from two bridging phosphonate groups and four water molecules, and each terminal metal is coordinated by two water and three oxygen atoms from three phosphonate groups. Adjacent metal ions are linked by one phosphonate group.

Please see **Lines 21-25/Page 10/MS** and **Figure S8/Page S36/SI**.

3) Mixtures of (*S*)-H₃L and metal salts are not soluble on common organic solvents, preventing studying their homogeneous catalytic activities. Interestingly, metal phosphonates crystallized from a 2:3 mixture of (*S*)-H₃L and the corresponding metal salts at r.t. are isostructural to Me₂L-Mg/Mn/Cr, as revealed by cell parameter determinations

Please see **Lines 25-28/Page 10/MS** and **Table S1/Page S8/SI**.

Notably, the catalytic activities of metal phosphonates prepared directly from H₃L and metal salts at r.t. are comparable with the unactivated MOFs **1-Cr/Mg/Mn**.

Please see **entry 3 vs entry 5/Tables S2-S5/Pages S13-S16/SI**

(c) It is unclear what the catalytically active units in the MOFs are responsible for the asymmetrical reactions.

[Author reply]

To address this issue, more experiments were conducted:

1) To exclude coexistence of free phosphoric acid and Lewis acid in the MOF catalysts, 1-Cr/Mg/Mn were used to catalyze hydrogenation of quinoxalines with Hantzsch esters, which

can be catalyzed by brønsted acids and cannot be promoted by Lewis acids [*Chem. Eur. J*, **16**, 2688 (2010)]. It was found that, under identical reaction conditions, **1-Cr/Mg/Mn** cannot provide any product even at elevated temperature, but both (S)-Me₂L and (S)-H₃L containing free phosphoric acid groups provided the product in good to high yields. The result suggests there does not exist free phosphoric acid in **1-Cr/Mg/Mn**. Please see **Table S6/Page S17/SI**.

- 2) After completion of the reaction catalyzed by **1-Cr/Mg/Mn**, the supernatant was condensed, and ³¹P NMR showed that there was no free phosphoric acid in the solution. The results precluded the possibility of the reaction catalyzed by the leaked chiral phosphoric acid. Please see **Tables S2-S5 & Figures S1-S4/Pages S13-S16/SI**.
- 3) In the FT-IR spectra, the absence of any absorption around 2350 cm⁻¹ implies that there are no free P–OH groups in **1-Cr/Mg/Mn**, consistent with the single-crystal X-ray diffraction. [*Inorg. Chem*, **55**, 5180 (2016)]. Please see **Figure S11/Page S39/SI**.
- 4) The water and/or methanol molecules coordinated to the terminal M in the linear trimeric M₃ unit that could be removed by heating under vacuum to generate coordinatively unsaturated metal centers, which could play a role as Lewis acid sites in catalysis. So the terminal M may take charge of chiral induction [*Applied Catalysis A: General*, **358**, 249 (2009); *Chem. Soc. Rev*, **46**, 3134 (2017)].

Taken together, these results suggested that there were no free phosphoric acid moieties and the terminal M ions can act as Lewis acid sites to catalyze the reactions with high activity and enantioselectivity. It should be noted that terminal metal nodes in MOFs are well-known to act as Lewis acid catalysts [For example, *J. Am. Chem. Soc.* **132**, 14321 (2010), *J. Am. Chem. Soc.* **138**, 9089 (2016)].

(d) I strongly suggest the authors to examine the structures of the homogeneous catalysts used in the comparative experiments before they draw a conclusion about the confinement effect of MOFs on the enantioselectivities.

[Author reply].

- 1) As mentioned above, we got the single crystals from the homogeneous catalyst systems and determined their structures. Based on element Analysis, ICP-MS, FT-IR and X-ray diffraction, they were formulated as [Cr₃(Me₂L)₆(H₂O)₅][OH]₃·3H₂O, [Mg₃(Me₂L)₆(H₂O)₈]-4H₂O and [Mn₃(Me₂L)₆(H₂O)₈]-3H₂O.

Please see **Lines 19-25/Page 10/MS** and **Figure S8/Page S36/SI**.

2) To further understand the confinement effect of MOFs, more control reactions with only Me₂L, H₃L, or metal phosphonates were performed. In all cases, only racemic products were obtained with moderate to good yields.

Please see **Lines 28-30/Page 10/MS** and **Tables S2-S5/Page S13-S16/SI**.

3) In literature, all related homogeneous catalysts that can induced high enantioselectivities contained very **bulky substituents at the 3,3'-positions of BINOL backbone** (such as – anthryl and -2,4,6-triisopropylphenyl), which can increase the steric interactions between the substrates and catalysts. In contrast, our reported ligands have a more flexible BIPNENOL backbone with less **bulky tert-butyl** groups at 3,3'-positions, which may be responsible for their inability of asymmetric induction in the reactions.

In the MOFs, chiral phosphono-carboxylate groups of 1,1'-biphenol, together with the metal ions and phenyl rings create a porous microenvironment, which may be responsible for the high degrees of catalytic activity and selectivity, by concentrating reactants and providing additional steric and electronic effects around the Lewis acid metal sites

Please see **Lines 30-37 (bottom, yellow)/Page 10 & Lines 1&2 (top, yellow)/Page 11/MS**.

(e) There are several minor suggestions. Variable temperature PXRD experiments should be also carried out to examine the thermal stabilities of the MOFs.

[Author reply].

Now, variable temperature PXRD experiments were conducted to study the thermal stabilities of the MOFs. The results showed that **1-Cr/Zr/Mg/Mn** were thermally stable up to about 375 °C.

Please see **Lines 27&28/Page 6/MS** and **Figure S13/Page S41/SI**.

(f) It is said that the catalyst amount is “4 mol% catalyst”. Does it mean that 4 mol metal %?

[Author reply].

We are sorry for no describing clearly.

You are right. “4 mol% catalyst” means that 4 mol metal % (= 2 mol **MOF %**). In the old version, “4 mol% of molecular catalyst” = 2 mol MOF% (only two terminal metal ions are believed as active sites). Because there exist three metal ions in the formula of the MOF, we changed the loading of molecular catalysts to be 6 mol % (based on metal) for the comparison of the catalytic activities. In fact, for the homogenous catalyst, the catalytic results with 4 mol% loading are almost the same as those with 6 mol % loading. This point was corrected in the revised MS and SI. For clarity, we used “2 mol % MOF” in the revised MS and SI.

Please see **footnotes of Tables 1-4/MS, related parts in MS and SI (Pages S9-S12)**.

Considering that the catalyst amounts are not the same, the results may be inappropriate for the comparison of the catalytic activities of different catalysts

(g) In the contrast test, the homogeneous catalyst amount is 5 mol%. Considering that the catalyst amounts are not the same, the results may be inappropriate for the comparison of the catalytic activities of different catalysts.

[Author reply].

We now used 6 mol% (based on metal, = 2 mol MOF%) homogeneous catalyst to perform the control reactions for the comparison of the catalytic activities. As mentioned above, the catalytic results are almost the same as those obtained with 6 mol % loading.

Please see **footnotes of Tables 1-4/Pages 7-9/MS, as well as Tables S2-S5/Pages S13-S16/SI.**

Reviewer #2 (Remarks to the Author):

Comments:

Although intensive progresses have been achieved in the field of heterogeneous asymmetric catalysis by MOFs, the tune of steric and electronic effects while retaining isoreticular structures is challenging, Cui and coworkers developed an attractive strategy to construct isoreticular MOFs using chiral phosphoric acid containing ligand and different kinds of metal ions, and disclosed their excellent performances in the heterogeneous asymmetric catalyses, which might fit the criteria of publication in Nat.Comm., if the authors could properly rationalize the following aspects.

1. It is interesting to tolerate several kinds of metal ions with different coordination abilities in MOFs with isoreticular structures, what is the critical factor inside? What is the state-of-the art of this point? And is it possible to extend this strategy to the broader scope of combinations of other ligands and metal ions?

[Author reply].

Thanks a lot for the review's insights.

- 1) In this manuscript, we demonstrated that up to 16 different metal ions could be assembled with the same linker forming the isostructural MOFs. The formation of such a large family of isostructures may be ascribed to the following reasons. First, the SBU formed $[M_3(CO_2)_2(O_2PO_2)(solvent)_2]$ in these structures features one metal in the center with octahedral geometry, and two metals in both ends with tetrahedral geometries. It can be seen that all the 16 metals with octahedral geometries are very common in coordination chemistry. It is worth mentioning that Zr^{4+} and Ti^{4+} ions etc tend to have higher coordination numbers. However, the *t*-butyl groups from the linkers, which are very close to the SBUs, shield the metal ions, and thus make these metals with tetrahedral geometries possible. Second, the acetic acid (HOAc) in MOF synthesis coordinates to the metals to compensate the charges in several cases, thus metals with different charges could fit into this framework. Third, the flexibility of the linkers also provides some freedom to fit metals with different sizes into the same structure.
- 2) We saw isostructural MOFs with up to seven different metals in the literature. For example, MOF-74 $[M_2(2,5\text{-dihydroxyterephthalate})]$ type frameworks with divalent metal ions Mg^{2+} , Cu^{2+} , Zn^{2+} , Co^{2+} , Mn^{2+} , Ni^{2+} and Fe^{2+} have been reported [Yaghi et al. *J. Am. Chem. Soc.*, **127**, 1504-1508 (2005)], Yildirim et al. *J. Am. Chem. Soc.*, **130**, 15268–15269(2008)]. To the best of our knowledge, our MOF series is with largest number of different metals ever reported.
- 3) We believe this strategy could be extended to the broad scope of combinations of various metal ions with ligands by carefully designing the geometries and steric shield groups of the organic linkers.

Please see **Lines 8-24/Page 5/MS**

2. In Fig. 3a, some curves like "Pristine 1-Cr" showed a special dependence of adsorption volume to P/P₀, and higher adsorption was achieved under a medium value of P/P₀. Why? Is there any pressure response of the MOF structure?

[Author reply]

We re-studied N₂ sorption isotherms of the pristine **1-Cr** at 77 K and found the special dependence of adsorption volume to P/P₀ disappeared. The new curve was similar to those of the other three MOFs. The curve was updated.

Please see Figure 3a/Page 6/MS and Figure S18a/Page S46/SI

3. With respect of asymmetric allyboration of aldehyde catalyzed by 1-Cr, there might be different rationalizations of asymmetric inductions. Cause that the chiral phosphoric acid also could catalyze this reaction and give high ee value, see *J. Am. Chem.Soc.* 2012, 134, 2716–2722; of course, the combination of BINOL and Lewis acidic metal ion like Sn also could lead to asymmetric induction, see *J. AM. CHEM. SOC.* 2008, 130, 8481–8490. Thus, there might be at least two different modes: the first is the residual free phosphoric acid moieties without coordination to chromiums that might be responsible for the asymmetric induction, and the second possibility will be like the statement by the authors-induced by the Lewis acidity of the chromium centers with chiral environments around them.

The question is how to distinguish those two kinds of possibilities? And which one will be responsible for the good performance of the title catalysis in paper? And how does it work? If the second mode is more possible, which one among the three chromiums in a unit will be more likely to take charge of chiral induction?

[Author reply]

To address your question, more experiments were conducted:

- 1) To exclude coexistence of free phosphoric acid and Lewis acid in the MOF catalysts, **1-Cr** was used to catalyze hydrogenation of quinoxalines with Hantzsch esters, which can be catalyzed by brønsted acids and cannot be promoted by Lewis acids [*Chem. Eur. J.*, **16**, 2688 (2010)]. It was found that, under identical reaction conditions, **1-Cr** cannot provide any product even at elevated temperature, but both (*S*)-Me₂L and (*S*)-H₃L containing free phosphoric acid groups provided the product in good to high yields. The result suggests there does not exist free phosphoric acids in **1-Cr**. Please see Table S6/Page S17/SI.

-
- 2) After completion of the reaction catalyzed by **1-Cr**, the supernatant was condensed, and ^{31}P NMR showed that there was no free phosphoric acid in the solution. The results precluded the possibility of the reaction catalyzed by the leaked chiral phosphoric acid. Please see **Table S2 & Figure S1/Page S13/SI**.
 - 3) In the FT-IR spectra, the absence of any absorption around 2350 cm^{-1} implies that there are no free P–OH groups in **1-Cr**, consistent with the single-crystal X-ray diffraction. [*Inorg. Chem*, **55**, 5180 (2016)]. Please see **Figure S11/Page S39/SI**.
 - 4) The water and/or methanol molecules coordinated to the terminal Cr in the linear trimeric Cr_3 unit that could be removed by heating under vacuum to generate coordinatively unsaturated metal centers, which could play a role as Lewis acid sites in catalysis. So the terminal Cr centers may take charge of chiral induction [*Applied Catalysis A: General*, **358**, 249 (2009); *Chem. Soc. Rev*, **46**, 3134 (2017)].

Taken together, these results suggested that there were no free phosphoric acid moieties and the terminal Cr(III) ions can act as Lewis acid sites to catalyze the reactions with high activity and enantioselectivity. It should be noted that terminal metal nodes in MOFs are well-known to act as Lewis acid catalysts [For example, *J. Am. Chem. Soc.* **132**, 14321 (2010), *J. Am. Chem. Soc.* **138**, 9089 (2016)].

- 5) Based on the above results and reported literature, we proposed a possible catalytic cycle [*Chem. Eur. J*, **19**, 124 (2013); *Chem. -Asian J*, **8**, 2033 (2013); *Chem. Commun*, **46**, 1260 (2010)]. The cycle involves the following four steps:
 - (a) The water or methanol molecule coordinated to the terminal Cr was removed to form coordinatively unsaturated metal center **A**.
 - (b) Exchange of ally active specie from boron to Cr to form intermediate **B**.
 - (c) Reacting with aldehyde through **six-membered transition state C** in a γ -addition fashion to give **D**.
 - (d) Hydrolysis of **D** to get homoallylic alcohol.Please see **Figure S5/Page S18/SI**.

Reviewer #1 (Remarks to the Author):

The authors have addressed in a reasonable way my initial comments. Acceptance is recommended without further revision.

Reviewer #2 (Remarks to the Author):

In the revised manuscript of Prof. Cui and coworkers, our comments have been well responded. It is our pleasure to see the freshly added re-study and deeper discussion, especially that the mechanism was proposed and proved effectively, and the structure-effect-relationship was nicely described.

Thus, we believed that the revised version has achieved the criterion of Nat. Commun. and hereby suggest accepting it directly.

Moreover, we are looking forward to the further development of this fancy isorecticular MOF chemistry of universal significance!